# CNNs efficiently learn long-range dependencies

**Timo Lüddecke and Alexander Ecker**
Neural Data Science Group, Institute of Computer Science
University of Goettingen
timo.lueddecke@uni-goettingen.de
ecker@cs.uni-goettingen.de

## Abstract

The role of feedback (or recurrent) connections is a fundamental question in neuroscience and machine learning. Recently, two benchmarks [1, 2], which require following paths in images, have been proposed as examples where recurrence was considered helpful for efficiently solving them. In this work, we demonstrate that these tasks can be solved equally well or even better using a single efficient convolutional feed-forward neural network architecture.

We analyze ResNet training regarding model complexity and sample efficiency and show that a narrow, parameter-efficient ResNet performs on par with the recurrent and computationally more complex hCNN and td+hCNN models from previous work on both benchmarks.
Code: https://eckerlab.org/code/cnn-efficient-path-tracing

## 1  Introduction

The proper use of recurrent processing could be crucial to enable neural networks to solve inherently iterative problems such as understanding subway maps or searching for complex patterns. While evidence from neuroscience suggests that human perception heavily relies on feedback connections, most successful computer vision models follow the feed-forward paradigm.

In an effort to determine the limits where recurrence is necessary, Linsley et al. [1] introduced the pathfinder challenge. Subsequently, the cABC dataset following a similar paradigm was proposed by Kim et al. [2]. In both tasks, dotted paths must be traced in order to decide if two points are connected via a path or pertain to different paths. Each image contains two points along with multiple distractions, and a learner must make a single binary decision for each image. Building upon earlier work using variants of gated recurrent units [3, 1], Kim et al. [2] present a convolutional recurrent neural network (hCNN) featuring horizontal connections, and its extension td+hCNN, which features horizontal connections and top-down feedback. They report that recurrent architectures perform well on pathfinder and cABC, respectively. In contrast, they observe that commonly used feed-forward architectures including ResNets and U-Nets fail at the highest difficulty level despite requiring magnitudes more trainable parameters than the proposed RNNs. Based on this result, the authors diagnose "a computational deficiency of feedforward networks."

**Contributions**   In this work we demonstrate that, contrary to the claims of [1] and [2], recurrence is not necessary for these path-finding tasks and convolutional neural networks are capable of solving even the most challenging variants efficiently. We investigate the factors that enable ResNets to learn solving these tasks reliably. We found that the network width accounts for only a small fraction of performance and ResNets can be stripped off substantially, improving upon td+hCNN in terms of parameter efficiency and inference speed (the latter holds for all ResNets) while matching its performance.

2nd Workshop on Shared Visual Representations in Human and Machine Intelligence (SVRHM), NeurIPS 2020.

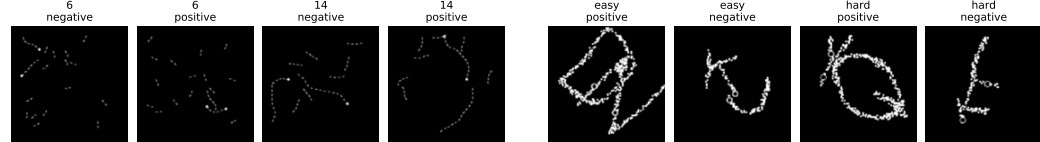

Figure 1: Four samples of different difficulty levels from the Pathfinder (left) and cABC datasets (right) each. Positive means a connections exists, while in negatives samples the dots are not connected.

**Related Work**  The question of when recurrence is required and how it is implemented is old and central to the fields of neuroscience and machine learning. For a long time, the primary choice for text processing were recurrent neural networks such as LSTM [4] or GRU [5], due to their ability to adapt the number of processing steps. However, it was found that temporal convolutional neural networks were able to solve these tasks [6]. Later, attention-based feed-forward architectures revolutionized the field [7]. For robotic path planning, CNNs were shown to work well, too [8].

## 2  Experiments

### 2.1  Pathfinder and cABC Datatset

The task of the *Pathfinder* dataset is to determine if two dots are connected by a curve (or path) or not (Fig. 1, left). Curves are dashed (i.e. composed of small segments) and can intersect. Both paths and dots are rendered into a single image. Hence the decision if a connection exists is a binary classification problem for which we can report an accuracy. Pathfinder consists of three datasets of varying difficulty, which is controlled by the path length: 6, 9 and 14 elements.

In follow-up work, Kim et al. [2] introduced the *Cluttered ABC (cABC)* dataset (Fig. 1, right), which shares the binary image classification format and the division into three levels of difficulty (easy, intermediate and hard). In contrast to Pathfinder, it has global semantics in form of letters, but is locally degenerate. The latter is due to paths being randomly sprinkled dots which follow a curve as an ensemble on a higher level. In order to follow paths here, dots must be grouped in a top-down manner. Indeed, Kim et al. [2] report that top-down feedback enables solving this task while they failed to train wide CNNs on this task.

### 2.2  Models

We focus our analysis on the ResNet model family, which has emerged as one of the predominant feed-forward CNN architectures. They use residual connections, which bypass computational blocks preventing vanishing gradients. Contrary to feedback connections, which induce information from higher levels of processing, residual connections are strictly feed-forward shortcuts. The original ResNet18 [9] designed for ImageNet [10] has around 11.2 million parameters.

#### 2.2.1  Narrow ResNet18 (nRN18)

We adopt the ResNet18 architecture [9] but reduce the number of channels, since the synthetic images used in this paper are less complex than the natural images ResNet was designed for. While the original ResNet18 contains blocks that have 64, 128, 256 and 512 feature channels, we reduce these numbers to 16, 32, 32 and 32 yielding around 120,000 trainable parameters while maintaining the network depth. Thus it matches the 142,000 parameters of hCNN [1]. We refer to this model as "narrow ResNet18," abbreviated by nRN18. We use two additional variants of nRN18 in some experiments: An extremely reduced network with channel numbers of 8, 16, 16 and 16, called super narrow ResNet18 (snRN18). This network has only around 30,000 parameters. Conversely, the larger-narrow ResNet18, lnRN18, has 24, 48, 48 and 128 channels.

---

[1]personal communication with the author [1]

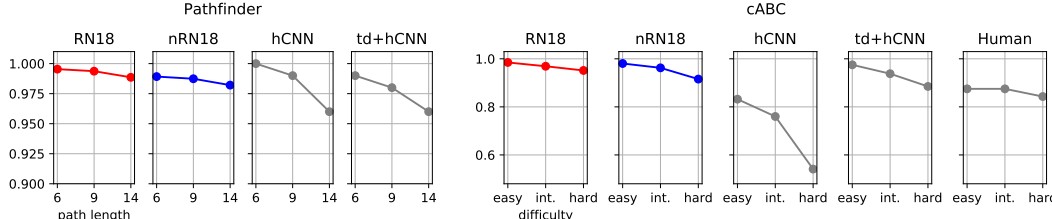

Figure 2: Accuracies of RN18 (red) and nRN18 (blue) on Pathfinder (left) and cABC (right). Scores in gray are adapted from [2].

## 2.3 Implementation

If not described otherwise, we use the Adam optimizer [11] to minimize the binary cross entropy loss with a learning rate of 0.001 and a batch size of 196. We train our models on 360,000 samples for Pathfinder and 144,000 samples for cABC, except for the analysis of sample efficiency (Fig. 6) and parameter efficiency (Fig. 5, left). Here we use 54,000 (Pathfinder) and 40,500 (cABC) samples in order to match the setup by Kim et al. [2]. We train for up to 80 epochs with an early stopping criterion after 7 epochs without improvement on the validation loss. For the efficiency experiments, the early stopping criterion is relaxed to 50 epoch and the validation accuracy is considered for stopping. Images are scaled to 150x150 pixels. We use the PyTorch [12] framework along with Numpy [**?** ] and OpenCV [13].

## 2.4 Results

Our main result is that both datasets – Pathfinder and cABC – can be solved with the same accuracy as models including recurrence and feedback [2] using a plain ResNet-18 (Fig. 2, red) or even a more parameter-efficient narrow ResNet-18 (Fig. 2, blue). On Pathfinder, nRN18 matches the performance of hCNN's best-of-5 run (Fig. 2, gray). On cABC, nRN18 even outperforms the state-of-the-art network td+hCNN [2] (hCNN fails on this task). Due to the different training set sizes, this comparison is not entirely fair, but demonstrates that ResNet can solve the benchmark very well given enough training samples.

These experiments show that feedback or recurrent connections are not required to solve the proposed tasks. In addition, feed-forward CNNs do not need more parameters to solve the tasks. nRN18 has approximately 120,000 parameters. The hCNN model used in [2] has slightly more parameters (144,000), while td+hCNN is substantially more complex (725,000 parameters). In fact, in section 2.4.1 we will see that we can use an even smaller model without sacrificing a lot of accuracy.

**Stability** In all other experiments we provide only point estimates of the accuracy rather than intervals due to computational demands. To assess how reliable these results are, we train nRN18 on Pathfinder-14 ten times. We make three observations (Fig. 3):

- Larger batch sizes are more stable.

- The larger RN18 performs better on cABC and is generally more stable than nRN18.

- Pathfinder seems to be more sensitive to batch size but in general easier to solve. The latter is suggested by a generally larger fraction of correctly classified samples in Pathfinder and a smaller gap between RN18 and nRN18.

**Training/Inference Speed** We found our models to train magnitudes faster than the public hCNN implementation (Tab. 1). The reason for computational complexity is that each hCNN-GRU cell carries out two convolutions with a filter size of 15x15 with stride 1 on a 150x150 input grid. Although the number of channels is fairly small (20), these convolutions in each time step require approximately 3.3 billion multiplications. The td+hCNN model, which is more complex than hCNN, uses 8 timesteps.

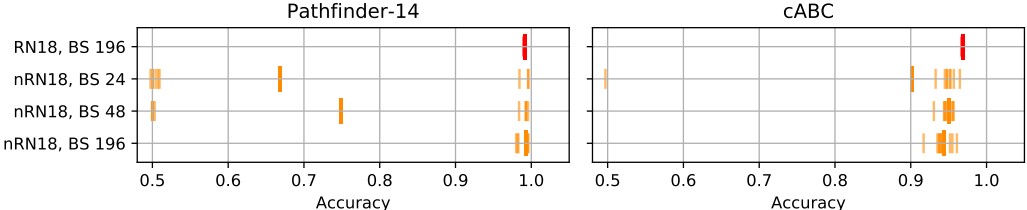

Figure 3: Training stability for RN18 and nRN18 for different batch sizes.

|  | RN18 | nRN18 | hGRU (15x15) |
|---|---|---|---|
| Training (360K 150x150 images) | 4.4 min | 3.0 min | 475 min |
| Forward (8 × 8 samples) | 0.64s | 0.23s | 34.2s |

Table 1: Approximate times required for training (measured on GPU) and forward passes (measured on CPU) on Pathfinder.

**Training Issues**   We found that small batch sizes (12 for RN18 and nRN18, 24 only for nRN18) can prevent the training on Pathfinder-14 from converging. Considering this in conjunction with the finding that small batch sizes tend to be unstable (Fig. 3), we recommend using large batch sizes. In addition, training progresses slowly at the beginning (Fig. 4). It might take more than seven runs over the entire dataset before the training loss begins to decrease significantly. Thus early stopping criteria must be chosen carefully.

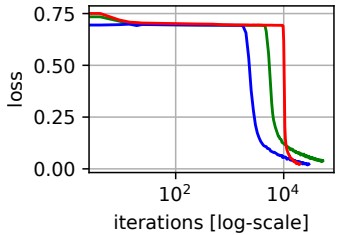

### 2.4.1   Model Complexity

A key claim of [1, 2] is that recurrence enables parameter-efficient models. Hence it seems natural to ask: How many parameters are actually needed by our ResNets?

Figure 4: Training loss of snRN18 (green), nRN18 (blue) and RN18 (red) on Pathfinder-14.

We find that by reducing the number of channels in each layer, we can train ResNet variants with smaller number of parameters than hCNN and td+hCNN to the same level of accuracy (Fig. 5, left).

Interestingly, the smaller the number of channels and thus parameters in a model, the more iterations are required for convergence (Table 2 and Fig. 5 right). This phenomenon can be explained with the lottery ticket hypothesis [14–16], which states that by randomly initializing the weights, subnetworks capable of partially solving the tasks emerge. As the chances of accidentally initializing a well-performing sub-network increase with model size, small models may have to learn more while larger models can rely on identifying a well-initialized sub-network.

|  |  | # iterations | | Acc | |
|---|---|---|---|---|---|
|  | params. | PF | cABC | PF | cABC |
| snRN18 | 0.03M | 53K | 22K | 97.3 | 89.3 |
| nRN18 | 0.12M | 29K | 10K | 98.1 | 91.9 |
| RN18 | 11.17M | 20K | 7K | 98.7 | 95.4 |

Table 2: Model complexity, training iterations and accuracies for ResNets: Smaller models require more iterations to train.

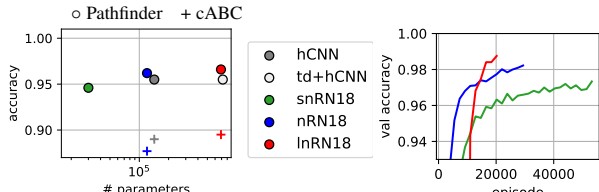

Figure 5: Left: parameter-performance trade-off when trained on 54K/40.5K samples (accuracy for hCNN on cABC is ∼54%). Right: validation accuracy during training on Pathfinder-14 (full dataset) for different models.

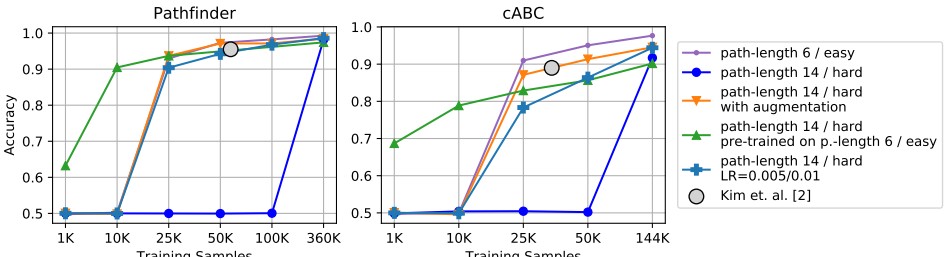

Figure 6: Classification accuracy for different training dataset sizes using nRN18.

### 2.4.2 Sample Efficiency

Sample efficiency expresses which performance (here in terms of accuracy) can be achieved with a fixed number of training examples, i.e. it measures how efficient a model makes use of the samples it sees. Choosing the right inductive biases, e.g. shift equivariance of convolutional layer for many image processing tasks, often improves sample efficiency.

A key argument of Kim et al. [2] is the high sample efficiency of their models as they use only 60,000 (Pathfinder) and 40,000 (cABC) samples. Hence, it seems natural to assess the sample efficiency of our ResNets models, too. To this end, we train each model for 25,000 iterations on a varying number of training samples. Validation error is computed (on 5,000 samples) every 100 iterations and early stopping with patience 20 is applied (number of validation runs without improvement after which the training is stopped).

First, we observe that the easy benchmark settings require fewer samples than the hard ones. In case of hard samples, nRN18 indeed seems to be more data-hungry than td+hCNN [2]. Thus, at first glance, these results seem to confirm the superior sample efficiency of the recurrent networks.

However, we found three simple ways to improve the sample efficiency of nRN18 and bring it on par with its recurrent competitors: (1) using a larger learning rate, (2) using a simple form of data augmentation by randomly flipping images along their horizontal and vertical axes (as done in [1]) and (3) using a form of curriculum learning by initializing the weights from a model trained on the easiest version of the respective dataset (Pathfinder: path-length 6; cABC: easy).[2]

Using a larger learning rate (0.005 for Pathfinder, 0.01 for cABC) improves sample efficiency substantially (at the price of training stability[3]). While this finding might seem surprising, it is consistent with previous research on batch normalization (BN): While Santurkar et al. [17] found BN to facilitate the optimization by smoothing both loss landscape and gradient, Bjorck et al. [18] argue the effects of improved generalization and faster convergence are due to the larger learning rates enabled by BN.

Furthermore, both other strategies to increase sample efficiency – data augmentation and pre-training – work well for nRN18, too (Fig. 6). The latter is particularly remarkable as it allows for very small training sets and shows that sample efficiency is to some degree a matter of choosing the right initializations. By contrast, the effect of augmentation sets in only above 25,000 samples.

In order to compare our results with Kim et al. [2], we train various ResNets using exactly the same training dataset sizes (Tab. 3). The additional lnRN18 model is a version of the narrow ResNet with larger channel sizes than nRN, yet it has fewer parameters than td+hCNN (and the normal ResNet18). In contrast to the previous experiment, here we use validation accuracy instead of loss as the stop criterion for both datasets.

We find that residual connections are not relevant for training on either dataset. This might be due to image statistics being different than natural images where residual connections work well. All our models come close to the performance of td+hCNN but only lnRN18 matches its performance, while

---

[2]For the pre-trained models, the validation interval is set to 50 instead of 100 iterations due to fast convergence.

[3]The reported numbers for Pathfinder at lr=0.005 for 100K and 360K samples were obtained in a second run after the first round of training did not converge. cABC uses a batch size of 512 at 144K samples. Note, Kim et al. [2] report best-of-five scores.

| model | Res. | channels | Accuracy | | parameters |
|---|---|---|---|---|---|
| | | | Pathfinder | cABC | |
| snRN18 | ✓ | 8-16-16-16 | 94.6 | 83.2 | 30,409 |
| nRN18 | ✓ | 16-32-32-32 | 96.3 | 87.7 | 119,697 |
| nRN18 | | 16-32-32-32 | 95.8 | 87.6 | 119,697 |
| lnRN18 | ✓ | 24-48-48-128 | 96.6 | 89.5 | 687,305 |
| td+hCNN [2] | | | 96 | 89 | ~720,000 |

Table 3: Comparison of sample efficiency using 54,000 and 40,500 training samples for Pathfinder and cABC respectively. The best accuracy over five runs is reported. Res. indicates if the model uses residual connections.

still being more parameter efficient. This demonstrates that recurrence is not a necessary component for sample-efficient or parameter-efficient learning on Pathfinder and cABC.

## 3   Conclusion

Our work shows that the Pathfinder and cABC benchmarks can be solved using parameter-effient and sample-efficient feedforward convolutional networks (ResNets), contrary to previous belief. There is no fundamental limitation that prevents CNNs from following paths through learning long-range dependencies, although training can be challenging due to instability of small batch sizes. We were able to use ResNets with only 30,000 parameters and found that a high sample efficiency can be attained through various means: large learning-rates, augmentation or pre-training. In a comparison with the recurrent td+hCNN [2] model our feed-forward ResNets were able to match its performance. The fact that our CNN architectures are identical in all experiments contradicts the claim by Kim et al. [2] that feedforward models must be specifically tuned for complex visual tasks. Despite these findings, we do believe that recurrence could be an essential ingredient for intelligent systems, particularly in tasks requiring a high degree of processing depth.

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
