# OpenReview forum: "CNNs efficiently learn long-range dependencies"
_NeurIPS.cc/2020/Workshop/SVRHM — SVRHM@NeurIPS Poster_

### Official Review · AnonReviewer1 · 2020-10-26
**Interesting and important reappraisal of the performance of ResNets on Pathfinder and cABC benchmarks**

**Rating:** 7
**Confidence:** 4

**Review:**

This paper demonstrates that benchmarks constructed to motivate the use of recurrent vision architectures can actually be solved by standard ResNets, with similar parameter- and sample-efficiency.

Strengths:

The submission provides compelling evidence both that Pathfinder and cABC tasks are too easy to be worth studying and that their use to motivate exotic architectures is misguided. This seems like a valuable contribution.

The work not only demonstrates that off-the-shelf ResNets can work, but also shows that they can be tuned to achieve similar parameter- and sample-efficiency.

Weaknesses:

The paper that introduced the Pathfinder dataset (ref. 1) evaluates ResNet-18 and claims that it achieves high accuracy at path lengths of 6 and 9 but fails at path length 14. I’m inclined to believe the results presented here, because I think it’s easier to make models fail than to make them work. Nonetheless, it would strengthen the work to know exactly how the training setup differs from those of refs. 1 and 2 of the submission that makes the ResNets here succeed whereas that previous work failed.

It’s unclear to me whether results regarding training time and stability in Figure 3, Table 2, and Figure 4 are meaningful. These findings are potentially sensitive to hyperparameters, but the authors do not appear tune the hyperparameters for different model or batch sizes. The authors seem to be surprised on L141 that tuning hyperparameters improves accuracy, but the value of hyperparameter tuning in deep learning, and particularly of tuning the learning rate, is well-established (see e.g. section 11.4 of the Goodfellow et al. deep learning book [1]). In addition, I wonder how much of the instability reported by the authors is related to the use of Adam with default values of the hyperparameters. Although refs 1 and 2 use Adam, most work involving ResNets uses SGD + momentum as in He et al. [2].

In Section 2.4.2, I'm not sure that the use of augmentation or pretraining is fair, since those methods effectively increase the training set size and could also improve sample efficiency of the baseline recurrent model. Nonetheless, the authors seem to achieve similar performance at the same number of training samples on Pathfinder simply by tuning the learning rate. On cABC, there is still a bit of a gap.

[1] Goodfellow, I., Bengio, Y., Courville, A., & Bengio, Y. (2016). Deep learning (Vol. 1, p. 2). Cambridge: MIT press.
[2] He, K., Zhang, X., Ren, S., & Sun, J. (2016). Deep residual learning for image recognition. In Proceedings of the IEEE conference on computer vision and pattern recognition (pp. 770-778).

---

### Official Review · AnonReviewer2 · 2020-10-28

**Rating:** 3
**Confidence:** 4

**Review:**

Summary:
This is a re-examination of recent works that have described a limited ability of feedforward models (ResNets as an example) to solve tasks which are designed to evoke perceptual grouping routines. They perform experiments over a range of training-set sizes not reported in the papers that introduced these tasks, and find the number of examples needed for successfully training ResNets. This number is much greater than what the feedback networks reported in those papers require. These results are paired with replications from the original papers, along with novel results on the stability of training for ResNets.

Strengths:

The authors replicated the results in Kim et al., 2020, which is great to see! Especially the dissociation between the importance of horizontal and top-down connections on Pathfinder vs. cABC.

Weaknesses:

The authors' argue a point about the ability of ResNets to learn Pathfinder and cABC. This is not disputed in Kim et al., 2020. In fact, in Figure S8 they report results from different ResNet parameterizations on Pathfinder and cABC,  much like the authors do here. Kim and colleagues show that *you can* change the architecture to do better, but that change doesn't hold for both tasks.

But the most critical point is that Kim and colleagues argue that feedback connections are important for *sample efficiency*. This is reinforced by the findings in the current paper. The authors do not describe a ResNet that can trace paths as efficiently as feedback networks.

In order to demonstrate that a ResNet can trace paths as efficiently or better than feedback networks, the authors must present a ResNet which can achieve comparable or better performance on the same or fewer training examples. The authors fail to demonstrate this.

---

### Official Review · AnonReviewer3 · 2020-10-30
**Great experiment and results, more discussion would be even better.**

**Rating:** 7
**Confidence:** 3

**Review:**

This paper seeks to understand the role of recurrence/feedback in neural system. While literature suggests that adding recurrence provides a significant boost to network performance for certain tasks, this paper aims to demonstrate that equivalent or better quality can also be achieved by a feed-forward network. This is a useful result that indicates that more analysis is necessary to fully characterize the necessity and utility of recurrence in neural networks.

The authors utilize the Pathfinder and cABC datasets for their experiment, which have been previously used to demonstrate the superiority of recurrence. They present results on variants of a ResNet18 model as the feed-forward network, finding that it is capable of beating the accuracy of prior state of art.

This work certainly causes the reader to question whether recurrence (with its training and representational complexity) is better than a carefully designed feed-forward architecture. The result is technically sound and supports the conclusion for the chosen architecture and datasets. Hence, I vote for acceptance.

However, a deeper dive into why residual networks perform unexpectedly good at these tasks would have helped the reader further. E.g.

* If we removed the residual connections, how much bigger would the network have to be for identical accuracy?
* Can recurrence help realize an even smaller ResNet?

Essentially, I am curious whether the improvements due to choosing a ResNet and due to using recurrence are orthogonal and additive. That would help shed some more light on the nature of these results. Regardless, it would be great to see this work published.

---

### Public Comment · ~Timo_Lüddecke1 · 2020-12-04
**Author Response**

We thank the reviewers for their valuable and helpful comments.

The main point of our paper is to show that vanilla ResNets with a reduced number of channels (width) solve the Pathfinder and cABC benchmarks as at least as well as complex recurrent models, which were previously considered to be necessary on these benchmarks. This means we obtain the same or better sample efficiency, parameter efficiency and accuracy.

**Sample Efficiency (Reviewer 2)**

Reviewer 2's main critique is that we do not address sample efficiency.
1. We explicitly assess sample efficiency in Section 2.4.2 and relate to the recurrent td+hCNN model by Kim et al. (2020). There we show that our nRN18 model becomes more sample-efficient by using larger learning rates. It matches the performance and sample efficiency of the much more complex td+hCNN.
2. Our paper discusses findings not only from Kim et al. (2020), which focused on sample efficiency, but also from the earlier work by Linsley et al.(2018) which used a non-sample efficient setting involving 900,000 training samples. These authors diagnosed a "computational deficiency of feed forward networks".
3. Kim et al. (2020) defines efficiency not only in terms of of samples but also in terms of parameters. We demonstrate a narrow ResNet18 (nRN18) which achieves the same performance with slightly fewer parameters than hCNN and only around 17% of the parameter budget of the td+hCNN model proposed by Kim et al. (2020).
4. The reviewer claims that the ability of ResNets to learn Pathfinder and cABC is not disputed by Kim et al. (2020). However, their conclusion clearly suggests this: "Our study also demonstrates a clear limitation of network models that rely solely on feedforward processing, including ResNets of arbitrary depths, ...".
5. Figure S8 of Kim et al. (2020) shows a wider ResNet (more parameters) which slightly increases scores in both tasks but does not match td+hCNN. We present a narrow ResNet (less parameters) which matches td+hCNN on both tasks.

In order to stress the point of sample efficiency we conducted an additional experiment using the same sample size as Kim et al. (2020) and report results in Table 3.


**Hyperparameters tuning works (Reviewer 1)**

The point we are trying to make here is that large learning rates increase the sample efficiency at the cost of stability. Hence, we cannot say that a larger learning rate is necessarily better.


**Use of augmentation and pre-training (Reviewer 1)**

We agree that comparing augmentation and pre-training with no augmentation and no pretraining is not fair.
However, the success of augmentation shows sample efficiency can be greatly improved by leveraging such a simple property as flip invariance.
The pre-training baseline highlights the importance of initializations and curriculum.
The main intention of the article is to gain insights of how feed-forward models can solve such tasks rather than beating baselines by a few percent.


**Performance without residual connections and recurrence + RN (Reviewer 3)**

We agree that this is an interesting follow up questions. We tested performance without residual connections and found it gave roughly the same performance. This shows  that residual connections are not important.
Adding recurrence into a ResNet is an interesting idea, too. However, it requires non-trivial model design decisions which are beyond the scope of this paper.

---

> ### Public Comment · ~Drew_Linsley1 · 2020-12-04
> **Feedback**
>
> Your title is funny, since Linsley et al and Kim et al show that some CNNs can learn Pathfinder. I think you should rethink this. Like prior work, you've shown that there are certain *types* of CNNs that can learn PF  — recurrent or otherwise — but it is far from trivial like standard CV benchmarks (MNIST, CIFAR, etc). Along those lines, recent work has shown that Pathfinder is a good benchmark for transformers: https://openreview.net/forum?id=qVyeW-grC2k This might be worth mentioning when discussing what kinds of models are able to solve this task.
>
> There are a number of changes which might explain the improvement over the original demonstrations of ResNets. What if HTD adopts the same improvements? Is there any reason to believe that the RNN will benefit less? Along those lines, you use larger batch sizes than either Linsley et al or Kim et al. Maybe you should investigate why this matters? Does it interact with batch norm in those models to smooth out the loss landscape enough to learn? What about training the models of Linsley et al., 2018 or Kim et al., 2020 with the 196 batch size instead of the 32 they used?
>
> Linsley et al., 2018 compared models when training for 1 epoch. Here you use at least 7 for some experiments and at least 50 for your sample efficiency experiments. So I would revise comparisons to that paper if I were you.
>
> Regarding Kim et al., I appreciate the efforts to replicate those results. Your work focuses on one element of that paper, the datasets, but misses its larger point, that recurrent models reveal a dissociation between horizontal vs. topdown feedback connections. Your work downplays that essential point, that these recurrent models can generate testable hypotheses for neuroscience and cognitive science. Indeed in that paper we found that the TD and H *decisions* correlated more strongly with humans on PF and cABC than feedforward models that solved those tasks. In other words, while CNNs can learn these tasks they do not seem to use the same decision strategies as humans.
>
> Overall, I appreciate the work, but I disagree with the conclusions.
>
> Drew Linsley and Junkyung Kim

---

> > ### Public Comment · ~Timo_Lüddecke1 · 2020-12-07
> > **Response**
> >
> > Hi Drew and Junkyung,
> >
> > thank you for your feedback on our article.
> >
> > We share your intuition that recurrence and feedback are important components of visual processing, and might be helpful for learning long-term dependencies. The experiments in your Kim et al. (2020) paper showing that errors of td+hCNN correlate better with human decisions are indeed very nice and we do not dispute them at all.
> >
> > However, we don't find evidence that feedforward CNNs are strained by Pathfinder or cABC as you stress in both of your papers. With relatively mild hyperparameter tuning, a single narrow ResNet can learn both tasks in equally parameter- and sample-efficient manner.
> >
> > In order to convey this argument more precisely we decided to change title and abstract.
> >
> > Timo

---

> > > ### Public Comment · ~Drew_Linsley1 · 2020-12-07
> > > **Thanks**
> > >
> > > Wonderful. Great work guys, and we're happy to see you were able to push these benchmarks. It is worthwhile to see if you can adjust the dataset generators to produce versions that strain your model, or testing your model on out-of-distribution/systematic generalization between versions of the datasets, which we found was especially difficult for purely feedforward models (Linsley et al., 2020. Stable and expressive recurrent vision models). For instance, generalizing to PF 20/25 after being trained on PF 14. Looking forward to seeing what comes next!
> > >
> > > Drew and Junkyung

---

### Decision · Program_Chairs · 2020-11-02

Accept (Poster)